# Sensitivity of Pathogenic Bacteria Strains to Treated Mine Water

**DOI:** 10.3390/ijerph192315535

**Published:** 2022-11-23

**Authors:** Catalina Stoica, Laurentiu Razvan Dinu, Irina Eugenia Lucaciu, Voicu Oncu, Stefania Gheorghe, Mihai Nita-Lazar

**Affiliations:** 1National Research and Development Institute for Industrial Ecology–ECOIND, 57-73 Drumul Podu Dambovitei, Sector 6, 060652 Bucharest, Romania; 2SC CEPROMIN S.A., 22 Decembrie 37A Boulevard, 330166 Deva, Romania

**Keywords:** conventional treated mine water, *Escherichia coli*, genotoxicity, growth inhibition, *Pseudomonas aeruginosa*, SOS chromotest

## Abstract

Mine water as a result of meteoric and/or underground water’s contact with tailings and underground workings could have an elevated content of metals associated with sulfate, often acidic, due to the bio-oxidation of sulfides. When entering aquatic ecosystems, the mine water can cause significant changes in the species’ trophic levels, therefore a treatment is required to adjust the alkalinity and to remove the heavy metals and metalloids. The conventional mine water treatment removes metals, but in many cases it does not reduce the sulfate content. This paper aimed to predict the impact of conventionally treated mine water on the receiving river by assessing the genotoxic activity on an engineered *Escherichia coli* and by evaluating the toxic effects generated on two Gram-negative bacterial strains, *Pseudomonas aeruginosa* and *Escherichia coli*. Although the main chemical impact is the severe increases of calcium and sulfate concentrations, no significant genotoxic characteristics were detected on the *Escherichia coli* strain and on the cell-viability with a positive survival rate higher than 80%. *Pseudomonas aeruginosa* was more resistant than *Escherichia coli* in the presence of 1890 mg SO_4_^2−^/L. This paper reveals different sensitivities and adaptabilities of pathogenic bacteria to high concentrations of sulfates in mine waters.

## 1. Introduction

Nowadays, the human population is facing a growing number of emerging environmental challenges, such as global warming, pollution and the contamination of air, water and soil, loss of biodiversity, overpopulation and natural resources depletion [1]. These challenges could have severe toxic impacts on biological organisms—including humans—which could result in diseases and premature deaths [2,3]. In addition to the above-mentioned environmental issues, mining activities, active or closed, are producing and have left significant ecological footprints and negative impacts on the environment, such as open pits, underground workings, tailings heaps containing high quantities of chemical compounds (including heavy metals, sulfides and polysulfides) and metals being leached and released in the mine water as metal sulfates, with harmful effects on receiving ecosystems, aquatic organisms and implicitly human health [4,5,6]. The acidity, heavy metals and metalloids are considered as the main contributors of the mine water’s toxicity, inhibiting the system metabolism, disturbing of carbon, nitrogen and organic matter cycles and reducing the microorganism biodiversity [7,8]. However, van Dam et al., 2014 [9] emphasized that in the absence of elevated concentrations of toxic trace metals, the major ions (or salinity) are a key source to a mine water’s toxicity, mostly due to osmotic stress [9]. Nevertheless, the complex interactions between the major ions and the large inter-species variability in responses to salinity make it difficult to relate toxicity to causal factors [9,10]. Sulfate (SO_4_^2−^) was identified as the main anion in acidic mine waters that could cause toxicity; however, SO_4_^2−^ was not the primary source of toxicity, as instead electrical conductivity was a better predictor of effects in case of mine waste waters [9].

By the conventional acid mine water treatment with calcium carbonate, calcium oxide as well as calcium hydroxide and oxidation with the air, heavy metal species are efficiently precipitated, being replaced by calcium (Ca^2+^) which associates with the existing SO_4_^2−^. Likewise, magnesium (Mg^2+^) will precipitate, related to the final pH [11]. Depending on the acid mine water’s initial SO_4_^2−^ concentration (if it is higher than 1500 mg/L), calcium sulfate can precipitate as gypsum, but the treated mine water will always contain SO_4_^2−^ associated mainly with Ca^2+^, along with Mg^2+^ and alkaline metals ions (including sodium (Na^+^) and potassium (K^+^)) [11,12]. The final SO_4_^2−^ concentration will result as a function of gypsum dissolution equilibrium in the aqueous system, but also on kinetic crystallization factors, including operational parameters. The final conventional acid mine water treatment effluent will have from approx. 1500 mg/L to metastable values up to 4000 mg SO_4_^2−^/L [12]. This SO_4_^2−^ load will be transferred to the surface water by the effluent discharge of the conventional acid mine water treatment. Efforts are made to address the mine water’s SO_4_^2−^ issue, both for closed mines and for new mining projects, at least to fulfill the legal discharge limits (e.g., from 300 to 1000 mg SO_4_^2−^/L); in spite of this, however, chemical and biological acid mine water treatment processes with advanced SO_4_^2−^ removal (to below 1500 mg SO_4_^2−^/L) are still under research and development [11,13,14,15]. The main treatment technology remains the conventional one, where the effluent water is a solution of calcium sulfate with a very low heavy metal content (i.e., below 1 mg/L) [12,16]. Other cations, such as the remaining Mg^2+^ as well as Na^+^ and K^+^, will be associated with SO_4_^2−^ in the effluent discharge, having relevant concentrations, but will be considerably lower than Ca^2+^ [11,13]. However, Na^+^, K^+^ and the anion chloride (Cl^−^) are not specific to non-ferrous ore-mine drainage [6,9,15].

Due to the fact that the impact of the treated acid mine water on the receiving water will depend on the flow rates/dilution, the chemical measurements are used to assess the quality of the treated acid mine water and some effects of the mine water or effluents discharging on the natural water bodies [14]. This approach will not describe the overall potentially negative effects; therefore a biological assessment is essential, including the organisms’ tolerance and adaptation-capacity evaluation [17].

Various freshwater organisms, such as primary producers—green algae (*Selenastrum capricornutum*)—as well as consumers (planktonic and benthic crustaceans—*Daphnia magna*—and *Heterocypris incongruens*, rotifers—*Brachionus calyciflorus*) and bacteria, were applied to analyze the toxic effects of chemicals such as surfactants, pharmaceuticals, pesticides, nitroderivates and heavy metals under Regulation (EC) no. 1907/2006-Registration, Evaluation and Authorization of Chemicals (REACH) [18], wastewater and sewage sludge from wastewater treatment plants from Romania [19], Danube and Danube Delta freshwater quality [20,21] or even mine water [22].

In addition, SO_4_^2−^ toxicity studies highlighted that among aquatic organisms, fish tended to have a much greater tolerance for SO_4_^2−^ than lower organisms [23]. Thus, bacteria may have the lowest tolerance to SO_4_^2−^, followed by algae, invertebrates and fish [24,25]. Moreover, a reduction of SO_4_^2−^ toxicity occurred in waters with increasing hardness, and Ca^2+^ plays an important role in this regard compared with Mg^2+^ through the formation of the ion pairs Ca^2+^ and SO_4_^2−^ mechanism [26,27]. Bacteria have the capacity to accumulate heavy metals as a result of the negative net charge of their cell envelope through a metabolism-independent passive or a metabolism-dependent active process that is determined by the absorptivity of the cell envelope and the ability to take up the heavy metal into the cytosol [28,29]. Along with heavy metals, salinity can directly affect freshwater organisms, including bacteria, either by (i) osmotic stress related to the combined elevated ion concentrations (e.g., a general salinity effect) and/or (ii) the toxicities of specific major ions or ion combinations [9]. It has been stated that a bacterial cell’s viability in cultures is influenced by extracellular salt effects [30]. The bacterial cells adapt to changes in osmolarity by changing their intracellular ionic strength [30,31] by two processes, such as uptaking and releasing Na^+^/K^+^ from and to the medium [32,33].

However, in vitro genotoxicity assays represent a simple, robust, rapid and cost-effective method of testing the toxicities of chemical compounds and their underlying mechanisms [24]. The test is based on linking the cell condition with the DNA damage induced by the toxic chemical compounds [26,27]. One advantage of using bacterial genotoxicity is the fact that it does not require a priori knowledge of toxicant identity and/or physical chemical properties; that is why unknown environmental samples are being tested [34].

Given that the impacts of the elevated SO_4_^2−^ concentrations are still poorly understood they are presumed mainly to be associated with the structures and functionings of freshwater ecosystems, and because the toxicity was often linked to osmotic stress rather than SO_4_^2−^ toxicity per se [35], the novelty of this study was to highlight the potential effects and the adaptabilities and sensitivities of freshwater pathogenic bacterial strains in laboratory conditions.

Moreover, considering the fact that chemical and biological mine water treatment processes with advanced SO_4_^2−^ removal are still under research and development [14], this study aimed to assess the potential toxic effect of a real acid mine water (high flow), conventionally treated, on an unimpacted river (low flow). The potential toxic effect was quantified by: (1) the growth inhibition of *Pseudomonas aeruginosa* and *Escherichia coli* Gram-negative bacterial strains, and (2) the genotoxic activity based on the mechanism of an engineered *Escherichia coli* using a blend of certain mine water, namely P2-Coranda, after treatment and the receiving Macris River water.

## 2. Materials and Methods

### 2.1. Study Area

The mining perimeter was located in the south-eastern part of the Romanian Metalliferous Mountains, being part of golder-quadrangle Sacaramb-Brad-Rosia Montana-Baia de Aries from about 20 km north-eastern of the Deva municipality. The ore exploitation, as well as the tailings dump perimeter, were located on the valleys of the Macris, Coranda and Baiaga streams. These streams are tributaries of Hondol river, which is the main collector in the area and a tributary of Mures River. From a physical–geographical point of view, the Hondol valley belongs to the typology of the Carpathian rivers that spring from low-mountain/piedmont areas [36]. The freshwater and the acid mine water study case (P2-Coranda) (45°59′50.1″ N; 23°00′08.9″ E) were based on the Certej mining perimeter, having a complex network of acid mine water sources (namely Baiaga, P1 + P2 Coranda) discharged without treatment into the Macris River (45°59′59.5″ N; 23°00′14.6″ E) (Figure 1). Certej is a sulfidation epithermal gold–silver deposit located in the southern part of the Apuseni Mountains in Romania [37].

The most three important ore deposits that were exploited in Certej’s catchment were Sacaramb, Hondol and Magura. After the construction of Coranda’s open pit in 1982, the exploitation continued in Coranda, as well as in Baiaga and Sacaramb [38]. In this study, Macris River water sampling was carried out upstream of the P2-Coranda discharge, being a non-impacted water.

### 2.2. Experiment Conditions

The acid mine water (P2-Coranda) was treated in laboratory by a conventional procedure as follows: precipitation with calcium hydroxide at pH = 9.5, with aeration, at 23 °C and with 30 min reaction time. The solids were separated by settling, resulting in the conventional treated mine water, namely effluent E1. The effluent E1 (2.4 volumes) was mixed with Macris River water (1 volume) to simulate the conditions that would occur at an aquatic system level by discharging the effluent, resulting Mix 1. The 2.4:1 ratio was the highest possible according to SC CEPROMIN SA flows-monitoring data [22].

The physico-chemical characteristics of mine water, river water and the simulated Mix 1 are detailed in Stoica et al. [22]. The chemical indicators were detected using appropriate standardized methods for metallic species, such as atomic absorption spectrometry [39,40,41], inductively coupled plasma mass spectrometry ICP-MS [42] and inductively coupled plasma optical-emission spectrometry ICP-OES [43]. The quantifications of the anions and groups IA and IIA’s cations were performed using SR EN ISO 10304-1 [44] and SR EN ISO 14911 [45], respectively. The mine water had a pH value of 2.69, electrical conductivity of 3.90 mS/cm, SO_4_^2−^ of 3532 mg/L, Ca^2+^ of 313 mg/L, Mg^2+^ of 311 mg/L and total dissolved solids (TDS) of 5360 mg/L. For the conventional treated mine water (E1), a pH value of 9.5 was identified, along with SO_4_^2−^, Mg^2+^ and TDS values that decreased down to 2998 mg/L, 43 mg/L and 4382 mg/L, respectively, at 3.58 mS/cm electrical conductivity [22]. The treatment with calcium hydroxide produced a 4-fold increase of Ca^2+^ concentration from 313 mg/L to 1281 mg/L. The metals were removed to low levels, e.g., Fe from 228 mg/L to 0.098 mg/L, Cu from 4.59 mg/L to <0.001 mg/L, Cd from 1.07 mg/L to 0.002 mg/L, Pb from 0.28 mg/L to <0.006 mg/L, Mn from 131 mg/L to 0.255 mg/L and Al from 167 mg/L to 1.84 mg/L [22].

The Macris River dilution water had a pH value of 7.65, an electrical conductivity of 0.20 mS/cm, SO_4_^2−^ of 26 mg/L, Ca^2+^ of 3.7 mg/L, Mg^2+^ of 5.7 mg/L, TDS of 118 mg/L and traces of heavy metals [22]. The Mix 1 water blend with a pH of 8.16 and electrical conductivity of 2.84 mS/cm had final concentrations of 1890 mg SO_4_^2−^/L, 715 mg/L Ca^2+^, 41 mg/L Mg^2+^ and 2476 mg/L TDS [22].

The chemical matrices of the two aqueous systems, P2-Coranda and Macris River, were characterized by low contents of alkaline metal ions and chloride.. The Na^+^/K^+^/Cl^−^ ratios, expressed as mg/L, were 65/22.5/61.3 for P2-Coranda, 9.6/1.6/1.3 for Macris River and 50.1/17.8/44.5 for Mix 1.

Overall, sharp increases of 73-fold for SO_4_^2−^, 193-fold for Ca^2+^, 21-fold for TDS and 14-fold for electrical conductivity were observed for Mix 1. Their toxic effects were further assessed in laboratory conditions by bacterial growth and genotoxic activity for this simulated downstream river water.

### 2.3. Bacterial Growth Inhibition

Gram-negative bacterial strains of *Escherichia coli* (ATCC 25922) and *Pseudomonas aeruginosa* (ATCC 27853) were purchased from ATCC (International Center for Authentification, Storage and Production of Microorganisms and Cell Lines, Manassas, Virginia, USA). The bacterial-growth-inhibition method was applied according to Nita-Lazar et al., (2016) [30]. Briefly, the bacterial strains were grown at 37 °C for 24 h on a solid nutrient medium (casein soya bean digest agar, Oxoid, Basingstoke, Hampshire, UK). One single colony of each bacterial strain was transferred to a sodium lauryl sulfate growing medium (LB) and incubated for 24 h at 37 °C under mild mixing (130 rpm) (New Brunswick Scientific Innova 44, Eppendorf, Hamburg, Germany). A bacterial density of 0.2 OD_600nm_ was incubated for up to 24 h in the presence of Mix 1 serial dilutions from 1890 mg/L (C1 100%), 945 mg/L (C2 50%), 472.5 mg/L (C3 25%), 236 mg/L (C4 12.5%) and 118 mg/L (C5 6.25%) SO_4_^2−^.

The inhibitory effect assessments of Mix 1 on the Gram-negative bacterial strains were expressed as LID (the lowest dilution at which no effect was observed), LOEC (lowest observed effect concentration) and EC20 or EC50 (the effective concentration that inhibited the growth of 20% or 50% of the bacterial strain’s population).

### 2.4. Genotoxicity Test

The genotoxicity activity was evaluated using an engineered *Escherichia coli* strain by SOS chromotest assay (Environmental Bio-Detection Products Inc., EBPI, Ontario, Canada). The detection and quantification of the genotoxic activity of Mix 1 were performed following the working protocol described in the SOS chromotest kit. All materials, including the necessary reagents, were found in the test kit. The lyophilized bacteria were rehydrated with a culture medium under aseptic conditions and were pre-incubated at 37 °C (Binder, Tuttlingen, Germany) overnight (approx. 14–16 h) until the optical density (OD) of the bacterial suspension ranged from 0.15 to 0.2. The OD of 0.215 at 600 nm was measured on a UV-VIS spectrophotometer (VWR International, Monroeville, PA, USA). Prior to the test’s start, a dilution was made with a culture medium of the bacterial suspension up to OD_600 nm_ = 0.05. The positive control was a 4-Nitroquinoline-1-oxide (4-NQO) in 10% dimethyl sulfoxide (DMSO) from 0.313 µg/mL to 10 µg/mL. The negative control was represented by 10% DMSO. A serial dilution of Mix 1 in 10% DMSO from 100% to 50%, 25%, 12.5%, 6.25%, 3.125% and 1.562% for the bacterial growth inhibition was tested to analyze its genotoxic effect. A total of 100 µL of bacterial suspension with 0.05 OD_600 nm_ was incubated in the presence of Mix 1 dilutions or controls for 2 h at 37 °C. The β-galactosidase (β-gal) synthesis was tested after 2 h of incubation in the presence of chromogenic substrate. In addition, the viability of the bacterial cells was confirmed using alkaline phosphatase and by measuring the absorbance at 420 nm in the sample well (A_420_S), in the blank well (A_420_B) and in the negative control well (A_420_N). A total of 100 µL blue chromogen solution was dispersed in the test plate and incubated at 37 °C for 90 min (Microplate reader, Clariostar, BMG Labtech, Ortenberg, Germany) for green color-spectrum development. Before and after incubation, the β-gal and alkaline phosphatase production was measured both at 600 nm and 420 nm, respectively. The growth factor (G), the activity of β-gal, the induction ratio (IF) of the SOS system and the cell survival rate (SR%) were calculated according to SOS chromotest assay.

The induction ratio (IF) was calculated according to Equation (1):IF = β-gal/G(1)

Initially, the growth factor (G) was determined using the absorbance values measured at 420 nm according to Equation (2).
G = (A_420_S − A_420_B)/(A_420_N − A_420_B)(2)

Subsequently, the activity of β-galactosidase was calculated based on the absorbance values measured at 600 nm, according to Equation (3):β-gal = (A_600_S − A_600_B)/(A_600_N − A_600_B)(3)
where:

A_420_S—the absorbance value measured at 420 nm in the sample well;

A_420_B—the absorbance value measured at 420 nm in the blank well (line H);

A_420_N—the averaged absorbance value measured at 420 nm in the negative control well;

A_600_S—the absorbance value measured at 600 nm in the sample well;

A_600_B—the absorbance value measured at 600 nm in the blank well (line H);

A_600_N—the averaged absorbance value measured at 600 nm in the negative control well.

In addition, the cell survival rate (%) was calculated to validate the results, as follows Equation (4):Survival rate (SR%) = OD_420_, sample/OD_420_, negative control × 100(4)

All of the OD measurements were calibrated based on the negative control, DMSO-without tested samples.

### 2.5. Statistical Analysis

The results were expressed as the average of three replicates (*n* = 3) ± standard deviation (SD). The experimental data was interpreted using ANOVA one-way, depending on the *p*-value. The *p*-value was considered insignificant for *p* > 0.05, significant for *p* < 0.05 and very significant for *p* < 0.01. The inhibition effects were estimated using regression the Hill model.

## 3. Results

### 3.1. Bacterial Growth Inhibition

The inhibition of bacterial growth was evaluated based on the optical densities (OD)–measured at 600 nm–after 24 h incubation at 37 °C and 130 rpm in the presence and absence of Mix 1 for all of the test solutions, as follows: C1-100%, C2-50%, C3-25%, C4-12.5% and C5-6.25%. The OD values at 600 nm after 24 h of exposure in the case of *Pseudomonas aeruginosa* was 1.097 from an initial OD-0 h of 0.230 ± 0.013, and for *Escherichia coli* it was 1.300 from an initial OD-0 h of 0.273 ± 0.009.

The results show an exponential bacterial growth after 24 h of incubation, both in the case of *Pseudomonas aeruginosa* (Figure 2a) and *Escherichia coli* (Figure 2b), although the growth curve was more robust in the first hour of contact in case of *Escherichia coli* (up to a OD_600 nm_-1 h of about one absorbance) compared with *Pseudomonas aeruginosa* (up to a OD_600 nm_-1 h of about zero point six absorbance). After 24 h of incubation, *Pseudomonas aeruginosa* registered higher bacterial densities compared with *Escherichia coli*. This bacteria behavior could be explained through an adapted response to the toxic pollutant in a time-dependent manner.

A decrease of the optical density was observed in case of the highest test solution concentration (C1 = 100%). The conventional treated mine water mixed with Macris River water (ratio 2.4:1) with a SO_4_^2−^ concentration of 1890 mg/L did not significantly influence the growth of either the *Pseudomonas aeruginosa* (*p* > 0.05, *p* = 0.43) (Figure 2a) or *Escherichia coli* strains (*p* > 0.05, *p* = 0.36) (Figure 2b), respectively. Even so, a decrease of 5% was observed in the case of *Pseudomonas aeruginosa* and of 47% in the case of *Escherichia coli* compared with the control at 1 h after incubation in the presence of Mix 1. After 24 h of incubation, the differences in optical densities between Mix 1 100% and the control were maintained, but the case of *Pseudomonas aeruginosa* was more accentuated of about 20%, while in the case of *Escherichia coli*, a growth recovery was observed (15%). No growth-inhibitory effect of *Pseudomonas aeruginosa* in the presence of Mix 1 for any of the tested solutions over the 24 h contact time was detected (Figure 2a). Instead, the *Escherichia coli* strains showed growth inhibitions in the dilution domain of Mix 1 of 25% to 100%. The growth of *Escherichia coli* was inhibited after 1 h of incubation in the presence of the 100% concentration test solution (Mix 1) (Figure 2b), decreasing by almost 7% after 4 h incubation time (94% after 1 h, 87% after 2 h, 881% after 3 h and 75% after 4 h). No growth inhibitions of *Escherichia coli* were recorded for the 12.5% or 6.25% tested solutions. It was observed that the growth inhibition (%) of *Escherichia coli* decreased proportionally with the decrease of the tested solution’s concentration. This fact could be associated with the expression of different genes involved in, e.g., sulfate transport as an adaptation mechanism [46].

The bacteria growth-inhibition values measured indirectly through the optical densities, LID and LOEC are estimated in Table 1 and Table 2.

The LID and LOEC values estimated for the tested bacteria revealed a dilution of Mix 1 in the range of 6.25 to 25% (118 mg SO_4_^2−^/L to 472.5 mg SO_4_^2−^/L), qualified as the highest concentrations with low-to-zero effect (Table 1).

The EC20 and EC50 values estimated for the tested bacteria revealed a dilution of Mix 1 of more than 50% in the range of concentrations from 52.57 to >100% for *Pseudomonas aeruginosa* and from 66.83 to >100% for *Escherichia coli* (Table 2). The obtained values of Mix 1 are qualified as non-toxic for tested bacteria, especially for *Pseudomonas aeruginosa*. In case of *Escherichia coli,* the EC50 values could be estimated after 1 h of exposure (71.31%), where the inhibition effects were >50% of the tested concentration. The estimated toxic concentrations were >1000 mg/L after 24 h exposure. An acute effect was observed after the first hour of exposure following by an adaptability and tolerance behavior of the tested bacteria.

Generally, the toxic values of EC50 were estimated to be >100% (>1890 mg SO_4_^2−^/L) for both of the target bacteria. The effective concentration at which inhibitory effects were recorded for 50% of the *Escherichia coli* bacterial population (EC50) was of 71.31% after 1 h of exposure time. The concentration-optimization curve for the SO_4_^2−^ ions against *Escherichia coli* growth-inhibition is depicted in Figure 3.

Considering the initial concentration of 1890 mg SO_4_^2−^/L detected for Mix 1 and EC50 after 1 h of incubation, we assume that a concentration of 1348 mg SO_4_^2−^/L can induce a growth inhibition of the *Escherichia coli* population (Figure 3).

### 3.2. Genotoxicity Assay

The genotoxic activity of Mix 1 on an engineered *Escherichia coli* strain was confirmed quantitatively by measuring the cell survival using alkaline phosphatase at 420 nm as well as the level of synthetized β-gal at 600 nm, respectively. Qualitatively, the visual analysis was reported on the positive control (4-NQO)’s color change as well as the optical density. The OD of the 4-NQO solution in 10% DMSO ranged between 1.618 (in the case of the highest concentration, 10 µg/mL) to 0.513 (in the case of the lowest concentration, 0.313 µg/mL). The results show that an OD > 0.78 becomes genotoxic; this OD value was induced by 0.625 µg 4-NQO/mL being the genotoxicity threshold in terms of concentration.

Quantitatively, the genotoxic activity of Mix 1 on *Escherichia coli* was quantified based on the growth factor (G), the activity of β-galactosidase (β-gal), the induction ratio (IF) of the SOS system as well as the survival rate (SR%) (Table 3).

The IF values of the SOS system obtained for the Mix 1 sample were related to the threshold IF values of 1.5–2.0, as stated in the SOS chromotest protocol (SOS chromotest, version 6.5, EBPI, 2016) [18]. For Mix 1, the induction ratio ranged between 1.09 (C1 100%) and 0.94 (C7 1.56%), decreasing proportionally with the sample dilution. Nevertheless, no genotoxic activity of Mix 1 (freshwater used for dilution of conventional treated mine water) was observed, although the highest induction ratio of 1.09 was calculated for the 100% concentration.

Furthermore, the cell survival rate was calculated to validate the results. Considering that a viability rate of 80% is necessary to confirm positive results, in the case of the Mix 1 water sample tested in this study, the SR (%) values confirmed a cell viability higher than 80% for all of the tested solutions, such as SR 100% = 99.6%; SR 50% = 100%; SR 25% = 93.7%; SR 12.5% = 97.9%; SR 6.25% = 99.7%; SR 3.125% = 94.2% and SR 1.562% = 99.5% (Table 3), which were confirmed also by qualitative results (optical density of 4-NQO variation).

Even if 1348 mg SO_4_^2−^/L possibly induces *Escherichia coli* growth-inhibition in the first 1 h of incubation time, no genotoxicity on the engineered strain was recorded. Moreover, the results on survival rate were confirmed by the growth-inhibition tests.

## 4. Discussion

A ubiquitous, free-living bacteria with pathogenic potential (*Pseudomonas aeruginosa*) and a facultative, anaerobic, non-spore-forming motile rod, commonly a resident of mammals intestines (*Escherichia coli*) [47], as well as the genotoxicity activity of an engineered *Escherichia coli* strain, were analyzed for their responses against conventional treated mine water mixed with freshwater in a ratio of 2.4:1 with the major pollutant SO_4_^2−^ of 1890 mg/L.

The Gram-negative bacterial species *Pseudomonas aeruginosa* and *Escherichia coli* were selected due to their abilities and developed resistance mechanisms for counteracting chemicals and/or environmental stress [30,48,49]. No significant effects were observed in the cases of the growth-inhibition tests both in the case of the *Pseudomonas aeruginosa* and the *Escherichia coli* strains. Both of the tested bacteria strains in the presence of Mix 1 revealed that the growth-inhibition percentage decreased proportionally with the decrease of the tested solution’s concentration, this being observed clearly in case of *Escherichia coli*. In addition, the results suggest that *Pseudomonas aeruginosa* could be more resistant than *Escherichia coli* in the presence of 1890 mg SO_4_^2−^/L. *Pseudomonas aeruginosa* could be more resistant than *Escherichia coli* due to intrinsic resistance mechanisms, which include its low outer membrane permeability (around 100-fold lower than that of *Escherichia coli*) [50]. This is because whereas both *Escherichia coli* and *Pseudomonas aeruginosa* are Gram-negative bacteria, their inner membrane lipid ratios differ [51]. Within the membrane of *Pseudomonas aeruginosa,* three additional lipids are found as the foremost components [51]. Sulfate-reducing bacteria such as *Pseudomonas aeruginosa* have an important role in the remediation of mine wastes due to their capacity to reduce the acidic waste [52]. This bacterium exhibited a great capacity to adapt to saline media through metabolic mechanisms for sulfur cycling having contribution in metals’ mobilities in soils [53].

Brahmacharimayum and Kumar Gosh, 2015 [32], showed that the *Pseudomonas aeruginosa* RI-1 (JQ773431.1) strain isolated from a wastewater sample proved to be effective for reducing SO_4_^2−^ from a synthetic water at 30 °C and a pH of 7.00. The same study showed that 500 mg/L and 1200 mg/L of SO_4_^2−^ were reduced in 4 days and 8 days, respectively, in the presence of the *Pseudomonas aeruginosa* strain [32]. Lasier and Hardin, 2010 [54], considered that SO_4_^2−^ is less toxic than chloride and bicarbonate as a result of chronic toxicity tests performed on aquatic organisms (*Ceriodaphnia dubia*); furthermore, Meays and Nordin 2013 [55], considered that 8000 mg SO_4_^2−^/L had no toxic effect on bacteria, protozoa and ciliates. Moreover, *Escherichia coli*’s sensibility was reported by other authors. Kushkevych et al., 2019, explained that *Escherichia coli* have the ability to metabolize sulfate into hydrogen sulfide (H_2_S) and further into sulfur amino acids, but H_2_S can permeate through the cell membrane into the cytoplasm without specific receptors, adjust the intracellular pH, damage the DNA and denature primary proteins [56].

The elevated levels of pollutants resulting from a mine exploitation could have important impacts on bacteria diversities, densities and activities. Some studies using bacteria-specific cultures showed changes in microbial function [57,58,59]. Bacteria show diverse changes in response to heavy metal pollution, and could adapt through their interactions [60].

Regarding the genotoxicity results, the literature reported a similar test (Ames test) performed for the evaluation of the genotoxic potential of water impacted by mine pollution using the *Salmonella typhimurium* TA98 and TA100 strains, that showed no mutagenicity [61]. The freshwater mixed with conventional treated mine water appeared to have a less-toxic effect on the target bacteria, inducing the idea that bacteria’s community structure can tolerate mine pollutants in high concentrations.

## 5. Conclusions

The evaluation of the growth inhibition generated by freshwater mixed with conventional treated mine water (Mix 1) on two Gram-negative bacteria, as well as the analysis of the genotoxic activity of Mix 1 on an engineered *Escherichia coli* strain, were the aims of the present study. This case-study had the particularity of using real acid mine water (high flow), conventionally treated, on an unimpacted river (low flow). In this case, the pollutant aqueous system resembled a solution dominated by calcium sulfate, with low-to-moderate contents of Na^+^, K^+^ and Cl^-^, as well as heavy metals as traces.

The results point out that *Pseudomonas aeruginosa* was more resistant than *Escherichia coli* to the presence of 1890 mg SO_4_^2−^/L. *Pseudomonas aeruginosa*’s growth inhibition (%) decreased after 24 h of contact with the increase of the bacterial population. Effective concentration at which inhibitory effects were recorded for 50% of *Escherichia coli* population (EC50) corresponded to the dilution of 71.31% vol. of Mix 1. Thus, a concentration of 1348 mg SO_4_^2−^/L can induce an inhibition of *Escherichia coli*’s population growth. The resistance mechanisms of bacterial cells related to ions toxicity must be further studied.

Mix 1’s solution containing SO_4_^2−^ could have influenced the adaptabilities and sensitivities of the studied freshwater pathogenic bacterial strains.

Nevertheless, no genotoxic activity of freshwater mixed with conventional treated mine water on the *Escherichia coli* strain was induced, and cell viability was observed (SR > 80%).

Future research must be focused on the temporal and spatial monitoring of mine water contaminates, including SO_4_^2−^ and its associated ions. In addition, the long-term effects of SO_4_^2−^ pollution need to be evaluated in situ in relation to the structures and compositions of autochthonous microorganisms and macroorganisms at various trophic levels along freshwater systems.

Furthermore, the impacts of abiotic factors (pH, dissolved oxygen, salinity, organic maters and other pollutants) and climatic changes must be taken into consideration for mine water risk assessments.

## Figures and Tables

**Figure 1 ijerph-19-15535-f001:**
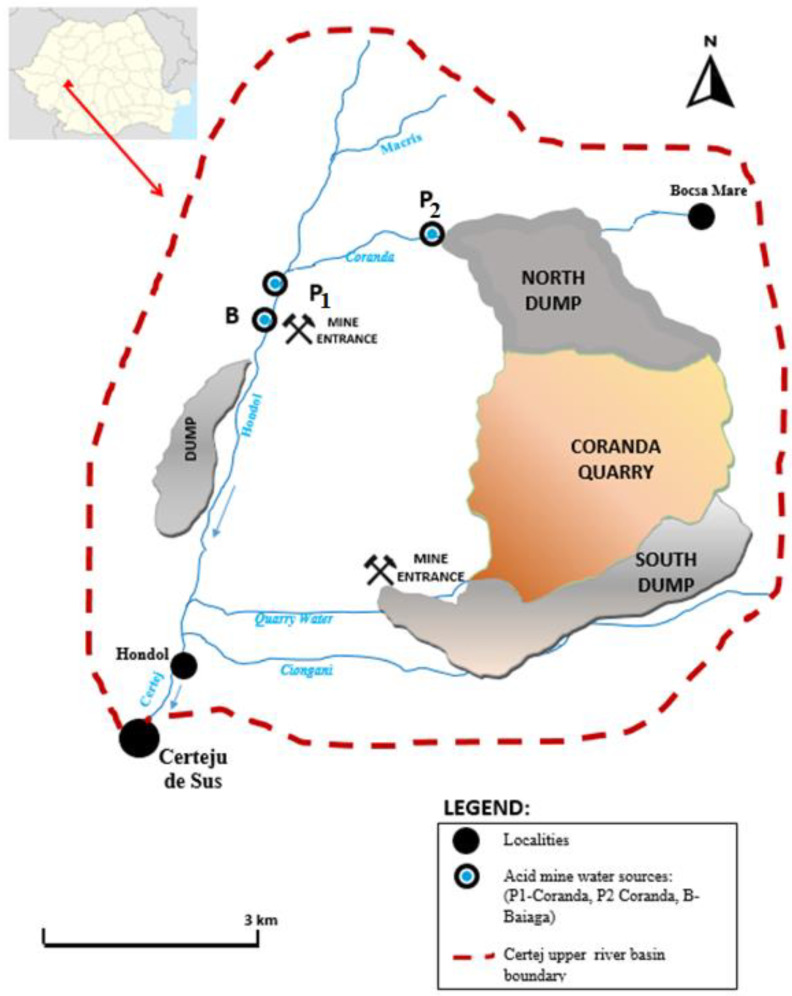
Map of the Certej area.

**Figure 2 ijerph-19-15535-f002:**
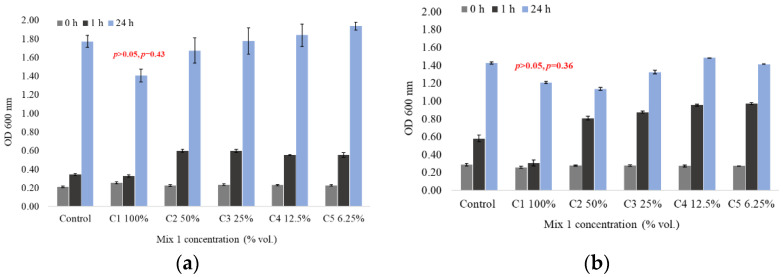
Gram-negative bacterial growth curve in the presence of Mix 1: (**a**) *Pseudomonas aeruginosa*; (**b**) *Escherichia coli* (average ± SD, *n* = 3). Control—bacterial strains in growth LB medium without Mix 1; Bacterial strains with Mix 1 serial dilutions from 1890 mg SO_4_^2−^/L (C1 100%), 945 mg SO_4_^2−^/L (C2 50%), 472.5 mg SO_4_^2−^/L (C3 25%), 236 mg SO_4_^2−^/L (C4 12.5%) and 118 mg SO_4_^2−^/L (C5 6.25%).

**Figure 3 ijerph-19-15535-f003:**
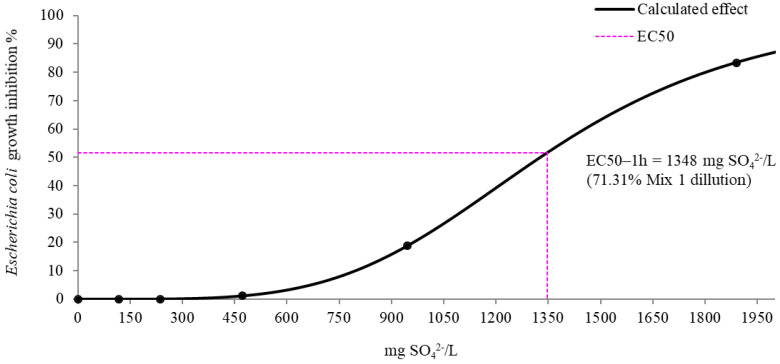
Estimation of EC50 after 1 h incubation of *Escherichia coli* in the presence of Mix 1.

**Table 1 ijerph-19-15535-t001:** LID and LOEC estimations for Mix 1.

* **Pseudomonas Aeruginosa** *	**Dilution (% Vol.) of Mix 1**	**Inhibition Effect%**
LID-1 h	<6.25	-
LID-24 h	6.25	0
LOEC-1 h	6.25	3.37 ± 1.23
LOEC-24 h	12.25	0.35 ± 0.02
** *Escherichia coli* **	**Dilution (% Vol.) of Mix 1**	**Inhibition Effect%**
LID-1 h	12.25	0
LID-24 h	12.25	0
LOEC-1 h	25	1.21 ± 0.04
LOEC-24 h	25	8.21 ± 1.02

Note: LID (the lowest dilution at which no effect was observed); LOEC (lowest observed effect concentration); results are expressed as average of 3 replicates ± SD; 6.25%—118 mg SO_4_^2−^/L, 12.25%—236 mg SO_4_^2−^/L and 25%—472.5 mg SO_4_^2−^/L.

**Table 2 ijerph-19-15535-t002:** EC20 and EC50 estimation for Mix 1.

* **Effect Endpoint** *	**Dilution (% Vol.) of Mix 1**	
	* **Pseudomonas aeruginosa** *	* **Escherichia coli** *
EC20-1 h	52.57 ± 4.35% (994 mg/L)	66.83 ± 6.32% (1263 mg/L)
EC20-2 h	64.96 ± 5.25% (1228 mg/L)	68.74 ± 8.36% (1299 mg/L)
EC20-4 h	65.00 ± 7.30% (1229 mg/L)	75.78 ± 10.51% (1432 mg/L)
EC20-24 h	94.74 ± 10.43% (1790 mg/L)	-
	** *Pseudomonas aeruginosa* **	** *Escherichia coli* **
EC50-1 h	>100% (>1890 mg/L)	71.31 ± 9.33% (1348 mg/L)
EC50-2 h	>100% (>1890 mg/L)	101.47 ± 12.56% (1917 mg/L)
EC50-4h	>100% (>1890 mg/L)	117.86 ± 11.43% (2227 mg/L)
EC50-24 h	>100% (>1890 mg/L)	>100% (1890 mg/L)

Note: EC20/EC50 (the effective concentration that inhibits the growth of 20%/50% bacterial strain population); results are expressed as average of 3 replicates ± SD; >100% (>1890 mg SO_4_^2−^/L).

**Table 3 ijerph-19-15535-t003:** The growth factor (G), activity of β-galactosidase (β-gal), induction ratio (IF) and survival rate (SR%) values of Mix 1 on the engineered *Escherichia coli*.

Mix 1 % Vol.	G	β-Gal	IF	Reference IF *	SR%	Reference SR *
**100%**	0.99	1.09	1.09	1.5–2.0	99.6	80%
**50%**	1.07	1.05	0.98		100	
**25%**	0.94	0.93	0.99		93.7	
**12.5%**	0.98	0.95	0.97		97.9	
**6.25%**	0.99	0.97	0.97		99.7	
**3.125%**	0.94	0.89	0.95		94.2	
**1.56%**	0.99	0.94	0.94		99.5	

Note: * Reference according to SOS chromotest protocol; the results represent the average value of the two independent experiments.

## Data Availability

Not applicable.

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
