# Peer review of "Sensitivity of Pathogenic Bacteria Strains to Treated Mine Water"

_ijerph, 2022, doi:10.3390/ijerph192315535_

Round 1
Reviewer 1 Report
In the manuscript entitled “Sensitivity of pathogenic bacteria strains to treated mine water” the authors reported the detection of the genotoxic activity of a conventional treated mine water after mixing with surface water on an engineered Escherichia coli as well as the evaluation of the toxic effects generated on two Gram-negative bacterial strains, Pseudomonas aeruginosa (ATCC 18 27853) and Escherichia coli (ATCC 25922). However, the manuscript contains some inconsistencies, and the results are not explained properly. Therefore, in my opinion, extensive revision with reformation is required before publication in this journal. The only some of the reasons are pointed out below.
1. In the introduction, the author did not justify why Sulphate ion was identified as principal ion that could cause the toxicity of the synthetic waters? Need reference or logically explain if it is the hypothesis of the study.
2. From the text, I do not see the novelty of the present work in relation to that known from literature data. The authors should report the advantages of the present work in relation to previously reported ones on similar issues.
3. Essential technical revision should be made throughout the manuscript as for example:
Page 1: “Chemical analysis of the environmental samples represents one of the shortest way to target specific compounds and pollution status, but not offer data about toxic effects to aquatic organisms.” => “Chemical analysis of the environmental samples represents one of the shortest ways to target specific compounds and pollution status, but not offer data about toxic effects to aquatic organisms.”
Page 1: “Both tailings and acid 34 mine drainage, respectively, contain and release high concentration of chemical com- 35 pounds (e.g., heavy metals)” => “Both tailings and acid 34 mine drainage, respectively, contain and release high concentration of chemical com- 35 pounds (e.g., heavy metals)”.
Page 1: “Moreover, complex interactions between the major ions (or salinity) are well-known as a key contributor to mine waters toxicity, because individual toxicity of a cation or anion may be masked or inseparably affected by the associated anion or cation [7].” => “Moreover, complex interactions between the major ions (or salinity) are well-known as a key contributor to mine water’s toxicity, because individual toxicity of a cation or anion may be masked or inseparably affected by the associated anion or cation [7].”
4. The figure 1 in the manuscript lacks adequate font size and some of the tests are completely unclear. It needs improvement to make readable the figure.
5. In the section 2.4, the OD of 0.215 at 600 nm was measured on the UV-VIS spectrophotometer. Why did you choose this wavelength should be justified. However, what do you mean by 0.215?
6. In section 3.1, It was observed that the growth inhibition (%) of Escherichia coli decreased pro- portionally with the decrease of the tested solution concentration-Why? No scientific explanation was found in the text, please explain it.
7. In section 3.1 (last lines): we assume that a concentration of 1348 mg/L SO4 2- can induce a growth inhibition of Escherichia coli population. To justify this please include the concentration optimization curve for SO4 2- ions against the growth inhibition.
8. In section 3.2, for genotoxicity assay, did you make any calibration curve? If so, it is suggested to include the calibration curve in this section.
9. In section 4, our results suggested that Pseudomonas aeruginosa could be more resistant than Escherichia coli in presence of 1890 mg/L SO4 2- , why? It is suggested to include the scientific explanation regarding this finding.
10. I miss the outcome or impact of this study including the drawback in the conclusion section.
Author Response
We would like to thank reviewer 1 for the detailed reading of our manuscript and constructive comments, which we have used extensively in improving this article!

Reviewer 2 Report
Please find the attached file.

Author Response
We would like to thank reviewer 2 for the detailed reading of our manuscript and constructive comments, which we have used extensively in improving this article!

Round 2
Reviewer 1 Report
I did not find the the concentration optimization curve for SO4 2- ions against the Escherichia coli growth inhibition in the revised manuscript.
Author Response
We would like to thank reviewer 1 for the comments and suggestions of the revised manuscript!
The concentration optimization curve for SO42- ions against the Escherichia coli growth inhibition was depicted in Figure 3 of the revised manuscript
(lines 326-327) of the revised manuscript.
Reviewer 2 Report
Please find the attached file.

Author Response
We would like to thank reviewer 2 for the time allocated for reviewing our manuscript which improved significantly the manuscript!
